# Evidence for Recent Polygenic Selection on Educational Attainment and Intelligence Inferred from Gwas Hits: A Replication of Previous Findings Using Recent Data

**Davide Piffer**

Department of Biology, Tuebingen University, 72074 Tuebingen, Germany; pifferdavide@gmail.com

**Abstract:** Genetic variants identified by three large genome-wide association studies (GWAS) of educational attainment (EA) were used to test a polygenic selection model. Weighted and unweighted polygenic scores (PGS) were calculated and compared across populations using data from the 1000 Genomes (n = 26), HGDP-CEPH (n = 52) and gnomAD (n = 8) datasets. The PGS from the largest EA GWAS was highly correlated to two previously published PGSs (r = 0.96–0.97, N = 26). These factors are both highly predictive of average population IQ (r = 0.9, N = 23) and Learning index (r = 0.8, N = 22) and are robust to tests of spatial autocorrelation. Monte Carlo simulations yielded highly significant p values. In the gnomAD samples, the correlation between PGS and IQ was almost perfect (r = 0.98, N = 8), and ANOVA showed significant population differences in allele frequencies with positive effect. Socioeconomic variables slightly improved the prediction accuracy of the model (from 78–80% to 85–89%), but the PGS explained twice as much of the variance in IQ compared to socioeconomic variables. In both 1000 Genomes and gnomAD, there was a weak trend for lower GWAS significance SNPs to be less predictive of population IQ. Additionally, a subset of SNPs were found in the HGDP-CEPH sample (N = 127). The analysis of this sample yielded a positive correlation with latitude and a low negative correlation with distance from East Africa. This study provides robust results after accounting for spatial autocorrelation with Fst distances and random noise via an empirical Monte Carlo simulation using null SNPs.

**Keywords:** educational attainment; polygenes; polygenic selection; IQ; GWAS

## 1. Introduction

Over the last decade, population geneticists have recognized that most traits are highly polygenic, and hence, have moved away from the study of genetic evolution using the single-gene, Mendelian approach, towards models that examine many genes together (i.e. polygenic models). Moreover, research into global variations in complex traits shows a significant amount of differentiation, for example in height [1], cardiovascular disease [2] and BMI [3,4]. Identifying polygenic adaptation is complicated by several factors. First, the presence of hundreds or thousands of genetic variants, each having a small effect, implies that signals of genetic differentiation are diluted, and that methods that target hard sweeps are not sensitive enough [5,6]. Second, identifying a number of SNPs which is sufficient to explain at least 5 or 10% of the total variance in a trait requires very large samples, and these have become available only recently (e.g. [7]). Third, environmental factors that differ across populations can influence the phenotype, hence masking genetic differentiation.

Signals of polygenic selection can be identified by various methods, such as correlation of allele frequencies [5,8–10] and the regression of population average of trait values on polygenic scores (PGS) [9]; these have been successfully applied to human stature [3,11,12] and cognitive abilities [9].

The goal of this paper is to test the predictive power of polygenic scores, independently of spatial autocorrelation and noise due to drift and migrations. The prediction is that the polygenic selection

model explains average population IQ better than a null model representing only drift and migrations. This implies that the frequencies of alleles with positive effect in the GWAS have different means across different populations.

A goal of this study is to replicate the effects found by Piffer [8,9], using evidence from new intelligence and educational attainment GWAS published to date.[8,9] analyzed educational attainment and intelligence GWAS hits and found a factor that was highly predictive of population IQ. The factor analytic method is based on the assumption that polygenic selection acts as a latent variable which accounts for commonalities among several genetic variants scattered across the genome [8]. The model also includes an error term due to measurement errors in the form of limited sample sizes, imperfect coverage or genetic drift, all of which act to increase noise.

Larger SNP sets become intractable with the factor analytic method; hence, the use of polygenic scores is usually preferred. These scores can be unweighted or weighted. Unweighted PGS are calculated as the sums of the number of effect alleles found in the genome for all trait-associated SNPs. A weighted PGS weights each trait-increasing allele by its odds ratio (for categorical traits) or β-value (for quantitative traits).

The chief limitation of polygenic scores is currently that the GWA studies were carried out on overwhelmingly European descent samples and this is responsible for a few issues, chiefly: 1) The GWAS will fail to capture population specific variants. This does not necessarily bias the PGS in favor of the reference group, as GWAS identify both negative and positive effect variants. For example, a recent GWAS carried out on Peruvians found a population-specific variant that reduces height by about 2.2 cm [13]. Since this variant is polymorphic only in populations of Native American descent, it would have been missed by a European-based GWAS, potentially leading to an overestimation (relative to Europeans) of the PGS for the Peruvian population. A similar scenario might happen with EA polygenic scores, where population-specific variants with negative or positive effects are missed in other populations, leading respectively to over and under-estimations of the non-European population polygenic score. However, since population specific variants can also have a positive effect, the effects will tend to cancel each other out, thus limiting the potential bias. Evidence suggesting that this is the case can be gathered from the polygenic score on height calculated using an European-based GWAS [14] which produced very low scores for Peruvians, the second lowest in the 1000 genomes samples (see section 3.4). 2)Since most GWAS hits are not causal (so-called "tag SNPs"), but are genetically linked with "true" causal variants, and because patterns of LD vary across populations (for example, Africans have on average much smaller LD blocks), this will reduce predictions for populations that are genetically distant from the GWAS sample.

Piffer (2017) [15](also see supplementary material) identified 9 genomic loci that were replicated across three of the largest GWAS of educational attainment (EA) [16–18]. The same 9 SNPs were successfully used to predict genetic differences in cognitive ability between ancient and modern samples [19].

In addition, the full set of 2411 genome-wide significant SNPs from the latest GWAS of educational attainment (henceforth, "EDU3") [7] will be employed.

Lee et al. (2018) [7] also reported a set of 127 putatively causal SNPs (posterior inclusion probability >0.9). These will be used to provide a polygenic score which is less subject to linkage disequilibrium-decay from a theoretical point of view (see discussion).

Average estimated population IQ will be used as the phenotype of interest and main dependent variable in the analyses. This choice can be justified by its privileged status in psychometric research and its robust genetic correlation with educational performance [20] and attainment [17]. Moreover, the GWAS hits identified by the three educational GWAS also predict general cognitive ability in their samples. A re-analysis of the Okbay et al. dataset revealed that the polygenic score also predicts general intelligence (3.6%) compared to 2% for the 2013 polygenic score [21]. The latest EDU PGS have been estimated to predict between 3.2% to 11–13% of variance in EA, depending on which set is used, and about 10% in general cognitive ability [7].

Height will be used as a control variable due to their highly polygenic nature which is consistent with cognitive ability, and because it is the anthropometric trait with the largest amount of GWA

studies available. Moreover, height is known to differ among human populations for genetic and environmental reasons.

Finally, socioeconomic variables will be added as predictors to model the effect of genetics and environment at the group level.

## 2. Materials and Methods

Lee et al. (2018) [7] reported 2415 SNPs reaching GWAS significance (P 5 × 10⁻⁸), and 2411 were found in 1000 Genomes. Weighted and unweighted PGS were computed for this sample ("EDU3").Since some of these SNPs are in LD, PGS was computed also for a set of LD clumped ("LD-free") SNPs (N = 1267) (for clumping algorithm, see Lee et al., 2018[7]).

One thousand genome VCF files were downloaded from Ensembl(ftp://ftp.1000genomes.ebi.ac.uk/vol1/ftp/release/20130502/), (The 1000 Genomes project consortium, 2015) [22]. These were converted to BED format using PLINK and into .csv using R. Polygenic score computation was carried out on R (code in supplementary files).

The Genome Aggregation Database (gnomAD) is the largest publicly-available genomic database to date [23], comprising 15,708 genomes from 8 populations. These are divided as follows: African/African American (4359); Latino/Admixed American (424); Ashkenazi Jewish (145); East Asian (780); Finnish (1738); Estonian (2297); North-Western European (4299); Southern European (53).

The SNPs frequencies were downloaded using the gnomAD browser v2.1.

Polygenic scores for height were calculated using GWAS hits from the largest meta-analysis to date [14]. Average height was obtained from the largest meta-analysis (NCD-RisC, 2016). However, for sub-populations within countries, separate estimates were retrieved from other sources (e.g. US Blacks *vs*. US Whites; Toscani in Italy; Chinese North *vs*. South).

Variables that were found to correlate with height in a previous study were included [24]. Since some of these variables can be expected to be correlated with cognitive development (via health, education and nutrition), they were included in this analysis.

For the socioeconomic variables, average protein consumption data was obtained from FAOSTAT.org (http://www.fao.org/faostat/en/#home ). The Human Development Index (HDI) ( http://hdr.undp.org/sites/default/files/2018_human_development_statistical_update.pdf ) computed by the UN and Infant Mortality (deaths/1,000 live births) was obtained from the World Factbook produced of the CIA(https://www.cia.gov/library/publications/the-world-factbook/)

Monte Carlo simulations were performed using a null dataset, consisting of a large sample (N = 2,411,000) of matched random unlinked SNPs (downloaded from 1000 Genomes, phase 3). Matching was carried out using SNPSNAP [25], by feeding the 2411 EA SNPs and setting LD r2 <0.25 (for EUR), with maximum allowed deviation for MAF = 5%. Unlinked SNPs were used in order to have a sample with independent observations.

The empirical value p = (r+1)/(n+1) was calculated, where r is the number of runs whose Pearson's correlation coefficient (r x population IQ) was higher than the one found using the actual (GWAS-derived) polygenic score; n = total number of runs. The corrected formula was provided by [26]. Fst distances were obtained from [9], calculated using Vcftools with 1000 Genomes, phase 3 data. Average population IQ estimates were obtained from Piffer (2015) [9]. Previously published scores were used also to guarantee that the values were not created post-hoc. In addition, recently published estimates of learning/education quality were included [27], based on performance on standardized tests of mathematics, reading, and science by 5-year age groups (from 5 to 19 years) for school-aged children.

Fst distances were used to partial out spatial autocorrelation, following the method outlined in Piffer (2015) [9] based on the Mantel (1967) [28] test. The method as it is commonly employed seeks spatial patterns of genetic variation by comparison of genetic distances, estimated by pairwise Fst, with geographic distances. In this study, the PGS distances were employed as the index of genetic distance for the EA-relevant polymorphisms, and the Fst distances were used as an index of "spatial" clustering due to phylogenetic relationships. Thus, the PGS distances were correlated to the Fst

distance matrix. Subsequently, pairwise population IQ differences were regressed on Fst and PGS distances to test if the observed relationship appears only because both variables are "spatially" structured by intrinsic effects (random drift or migrations) or if the two matrices are "causally" related, indicating positive directional selection or diversifying selection. Various forms of the partial Mantel test have been widely used in ecological settings using geographic distance and some environmental variable such as temperature [29] and its polygenic scores form was devised by Piffer (2015) [9] for polygenic evolution modelling.

HGDP-CEPH data were downloaded from SPSmart [30] and PGS were calculated on R after matching effect alleles (code can be found in supplemental files). Fst distances and distance from Addis Abeba for HGDP populations were obtained from (Handley et al., 2007) [31], and after removal of the non-overlapping samples, 49 matching populations were retained.

Statistical analyses were run using R (R CoreTeam, 2018) [32].

## 3. Results

### 3.1. Correlation between Polygenic Scores and Population Iq

The polygenic scores (Table 1) obtained from the latest GWAS and from Piffer's previous publications were highly inter-correlated, and they were highly correlated to an estimate of average population IQ (Figure 1).

**Table 1.** Polygenic scores and population IQ.

| Population | IQ | Learning | EDU3 (weighted) | EDU3 (Unweighted) | EDU 3 (Causal) | EDU2 | Piffer 2015[9] |
|---|---|---|---|---|---|---|---|
| Afr.Car.Barbados | 83 | 72 | −1.30 | 0.474 | 0.444 | −1.48 | −1.25 |
| US Blacks | 85 | | −1.28 | 0.475 | 0.445 | −1.07 | −1.05 |
| Bengali Bangladesh | 81 | 69 | −0.16 | 0.488 | 0.462 | 0.04 | 0.03 |
| Chinese Dai | | | 1.09 | 0.499 | 0.472 | 0.88 | 0.83 |
| Utah Whites | 99 | 89 | 0.76 | 0.497 | 0.480 | 0.40 | 0.53 |
| Chinese, Bejing | 105 | | 1.61 | 0.503 | 0.477 | 1.54 | 1.53 |
| Chinese, South | 105 | 95 | 1.42 | 0.501 | 0.476 | 1.32 | 1.32 |
| Colombian | 83.5 | 66 | 0.23 | 0.493 | 0.477 | −0.08 | 0.02 |
| Esan, Nigeria | 71 | 64 | −1.30 | 0.474 | 0.447 | −1.45 | −1.39 |
| Finland | 101 | 91 | 1.10 | 0.499 | 0.483 | 1.08 | 0.64 |
| British, GB | 100 | 88 | 0.59 | 0.495 | 0.482 | 0.78 | 0.68 |
| Gujarati Indian, Tx | | | 0.23 | 0.492 | 0.465 | 0.46 | 0.33 |
| Gambian | 62 | 65 | −1.30 | 0.474 | 0.439 | −1.56 | −1.57 |

| | | | | | | | |
|---|---|---|---|---|---|---|---|
| Iberian, Spain | 97 | 86 | 0.22 | 0.497 | 0.480 | 0.54 | 0.52 |
| Indian Telegu, UK | | 66 | 0.25 | 0.492 | 0.469 | 0.18 | 0.27 |
| Japan | 105 | 94 | 1.14 | 0.497 | 0.484 | 1.27 | 1.41 |
| Vietnam | 99.4 | 82 | 1.18 | 0.499 | 0.477 | 1.18 | 1.23 |
| Luhya, Kenya | 74 | 66 | −1.38 | 0.474 | 0.441 | −1.40 | −1.69 |
| Mende, Sierra Leone | 64 | 65 | −1.53 | 0.472 | 0.436 | −1.40 | −1.49 |
| Mexican in L.A. | 88 | 74 | −0.29 | 0.488 | 0.481 | -0.18 | 0.04 |
| Peruvian, Lima | 85 | 70 | −0.99 | 0.483 | 0.465 | -0.60 | 0.20 |
| Punjabi, Pakistan | 84 | 68 | −0.01 | 0.489 | 0.470 | 0.33 | 0.16 |
| Puerto Rican | 83.5 | 73 | 0.10 | 0.491 | 0.474 | −0.06 | −0.05 |
| Sri Lankan, UK | 79 | 75 | 0.05 | 0.490 | 0.466 | 0.13 | −0.09 |
| Toscani, Italy | 99 | 86 | 0.78 | 0.497 | 0.479 | 0.54 | 0.45 |
| Yoruba, Nigeria | 71 | 64 | −1.22 | 0.475 | 0.440 | −1.40 | −1.63 |

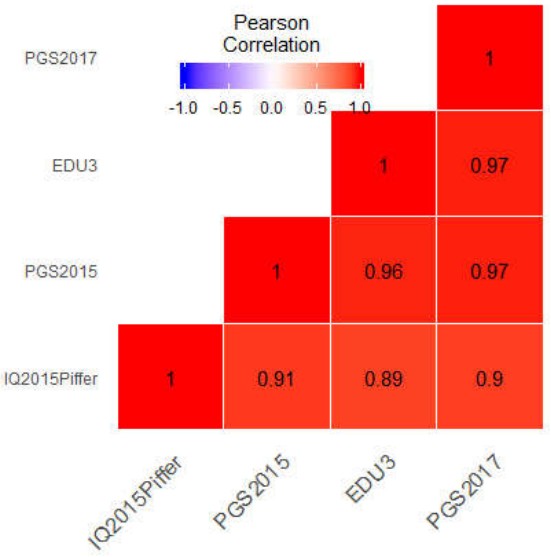

**Figure 1.**

Correlation matrix.

For the Lee et al. (2018) [7] GWAS, weighted and unweighted polygenic scores (EDU3) were almost identical (r = 0.98), and they had identical correlation with population IQ (r = 0.89) (Figure 2). Hence, the unweighted PGS will be used (Figure 3). In addition, Lee et al. (2018) [7] carried out a joint analysis of GWAS results on EduYears, cognitive performance, math ability and highest math using MTAG (multi-trait analysis of GWAS). The PGS computed from the EA MTAG derived SNPs was almost identical to the classical score (r = 0.997). MTAG also identified SNPs associated with cognitive

performance in a smaller sample (N = 257,841). The CP PGS (N = 1597) was highly correlated to the EA PGS (r = 0.925) and to population IQ (r = 0.861).

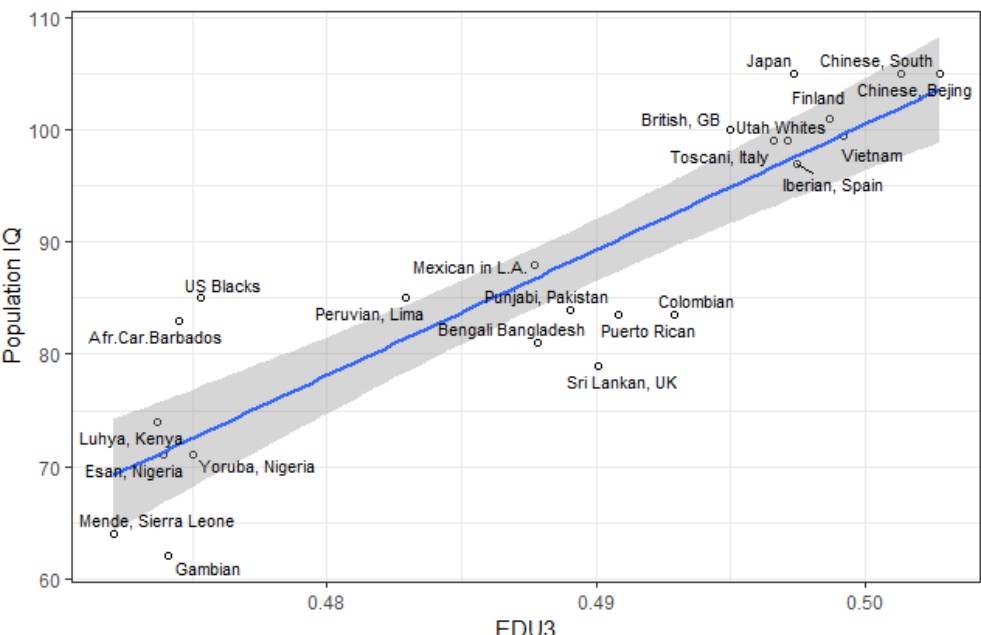

**Figure 2.** Correlation between EDU3 PGS and population IQ.

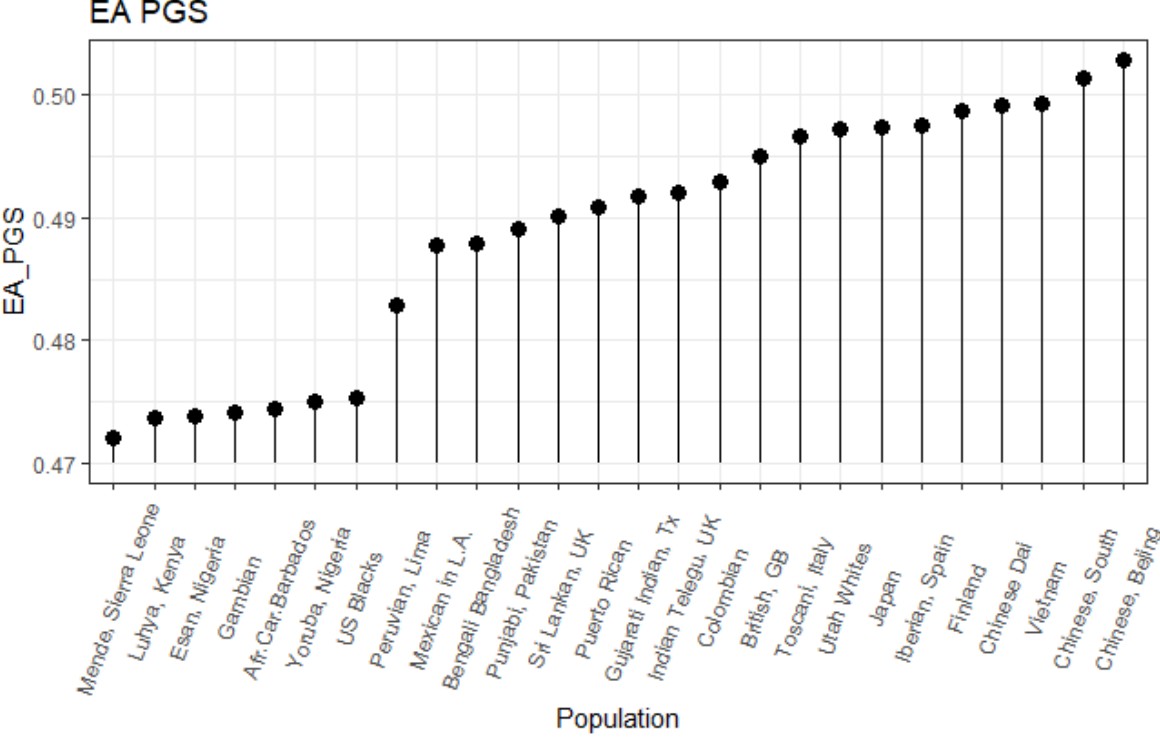

**Figure 3.** Polygenic score by population (GWAS significant hits, N = 2411).

As an additional check of their predictive validity, the correlation with Learning was computed (r = 0.797).

The putatively causal PGS was also highly correlated to population IQ (r = 0.82–0.85 for the weighted and unweighted PGS, respectively). The EDU3 PGS was negatively correlated to the difference between EDU3 and this score (r = –0.62). The PGS computed from the LD-clumped SNPs (N = 1220) had a similar correlation to population IQ (0.86), and it was highly correlated to the unclumped PGS (r = 0.98).

## 3.2. Monte Carlo Simulation

A Monte Carlo simulation was run using 943 PGS computed from groups of 2411 SNPs taken from the random dataset, with random assignment of one allele of each SNP to the "high-attainment" category. The average correlation between population IQ and the random polygenic scores was 0.329 (N = 943); this is shown in figure 4. The slightly positive correlation can be interpreted as an effect of spatial/phylogenetic autocorrelation. A test based on a Monte Carlo approach with null SNPs was carried out: the corrected (and more conservative) calculation of Monte Carlo p value, where p = r+1/n+1 (see Methods) was used, producing p = 0.001 (n = 943, r = 0.886). The polygenic score obtained from 9 quasi-replicated SNP yields p = 0.011 (n = 819, r = 0.88).

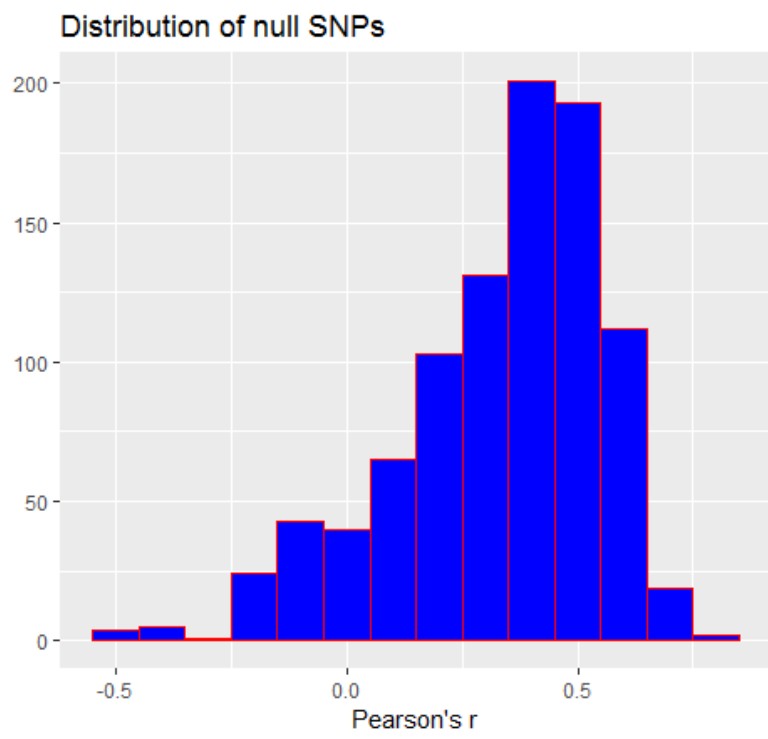

**Figure 4.** Correlation between null PGS and population IQ.

## 3.3. Controlling for Spatial (Phylogenetic) Autocorrelation

The presence of spatial autocorrelation in a dataset means that the cases are not independent, leading to an overestimation of degrees of freedom and, in the case of positive autocorrelation, a possible inflation in the correlation between two or more variables. The source of spatial autocorrelation in population genetics datasets is the similarity caused by admixture among neighboring populations, and the differences caused by random drift. Demonstrating that the alleles predict population-level differences in average phenotypic values above and beyond those predicted on the basis of migration, drift etc. provides evidence for a model of polygenic selection (Berg & Coop, 2014)[10].

The positive correlation between the two distance matrices (pairwise Fst distances and pairwise EDU 3 differences) was large (Mantel's r = 0.812, p = 0.001), indicating the presence of SAC. A

correlogram confirmed the trend for the correlation between Fst and EDU PGS distances to decrease with genetic distance (Figure 5)

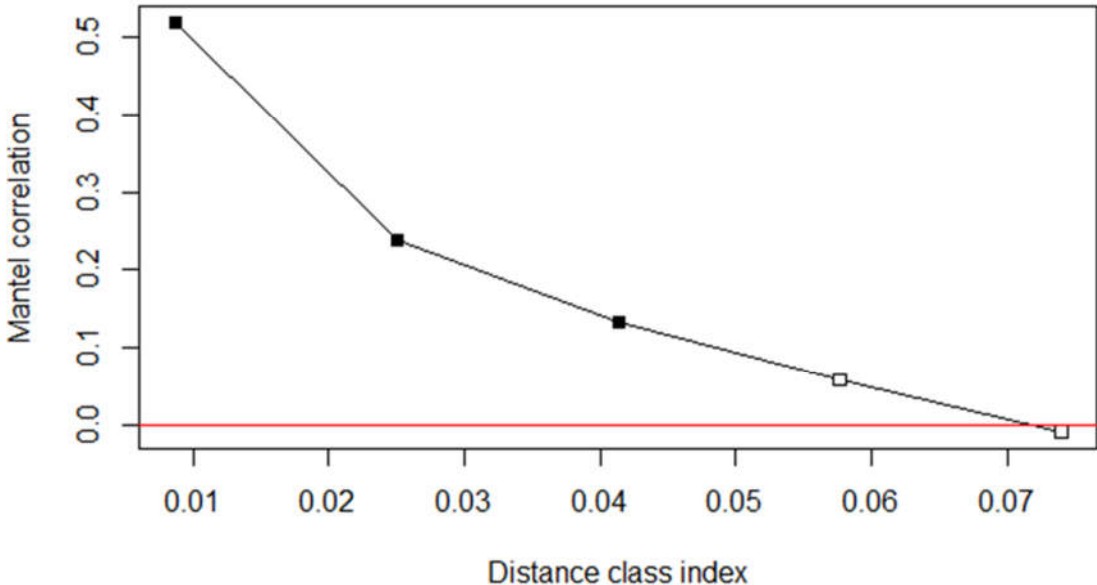

**Figure 5.** Correlogram (EDU 3 and Fst).

In order to partial out SAC, multiple regression was performed with pairwise PGS differences and Fst distances as predictors and pairwise population IQ differences as dependent variable. Significant models (respectively for 9 and 161 SNPs and EDU3) were obtained (Adjusted R-squared: 0.503, F-statistic: 128.5, $p < 2.2 \times 10^{-16}$), (Adjusted R-squared: 0.7251, F-statistic: 30.01, $p = 9.514 \times 10^{-7}$), (Adjusted R-squared: 0.510 ,F-statistic: 132.3, $p < 2.2 \times 10^{-16}$) (Table 2). Another way to remove SAC is via partial or semi-partial correlation: both yielded significant correlation coefficients (respectively, $r = 0.506$; $p = 7.85 \times 10^{-18}$ and $r = 0.409$; $p = 1.304 \times 10^{-11}$).

Using Learning instead of IQ produced significant results, with a significant model (Adjusted R-squared: 0.285, F-statistic: 46.94, $p < 2.2 \times 10^{-16}$).

Partial and semi-partial correlation were of similar magnitude (0.466, $p = 7.86 \times 10^{-14}$ and 0.443, $p = 1.59 \times 10^{-12}$).

**Table 2.** Multiple regression of IQ distances on Fst distances and GWAS hits distances

| Variable | Beta | T | Sig | VIF |
|---|---|---|---|---|
| *Model 1* | | | | |
| PS 9 distances | 0.524 | 9.024 | $<2 \times 10^{-16}$ | 1.713 |
| Fst distances | 0.250 | 4.305 | $2.4 \times 10^{-5}$ | |
| *Model 2* | | | | |
| PS 161 distances | 0.456 | 7.063 | $1.61 \times 10^{-11}$ | 1.912 |

| Fst dist | 0.274 | 4.241 | $3.14 \times 10^{-5}$ | |
| --- | --- | --- | --- | --- |
| *Model 3* | | | | |
| EDU3 | 0.680 | 9.290 | $<2 \times 10^{-16}$ | 2.761 |
| Fst dist | 0.045 | 0.615 | 0.539 | |
| *Model 4\** | | | | |
| EDU3 | 0.772 | 7.964 | $7.87 \times 10^{-16}$ | 3.026 |
| Fst dist | −0.324 | −3.344 | 0.00096 | |

\* Regressed on Learning index pairwise differences ("distances").

ANOVA

Frequencies of positive effect alleles were computed for one population from each of the 4 super populations (AMR was excluded because it contains heavily mixed samples). To assure independence between data points, the LD clumped subset was used and 1270 SNPs were found. The following values, shown in Figure 6, were obtained: Yoruba = 47.6%; Bengali = 49.5%; Finns = 50.7%; Chinese (South) = 50.5%. One-way ANOVA was carried out to test the significance of the allele frequency differences between groups. A model with trend towards significance emerged (F = 2.594; P = 0.0509). The p value floated around 0.05 depending on the choice of populations from each super-population.

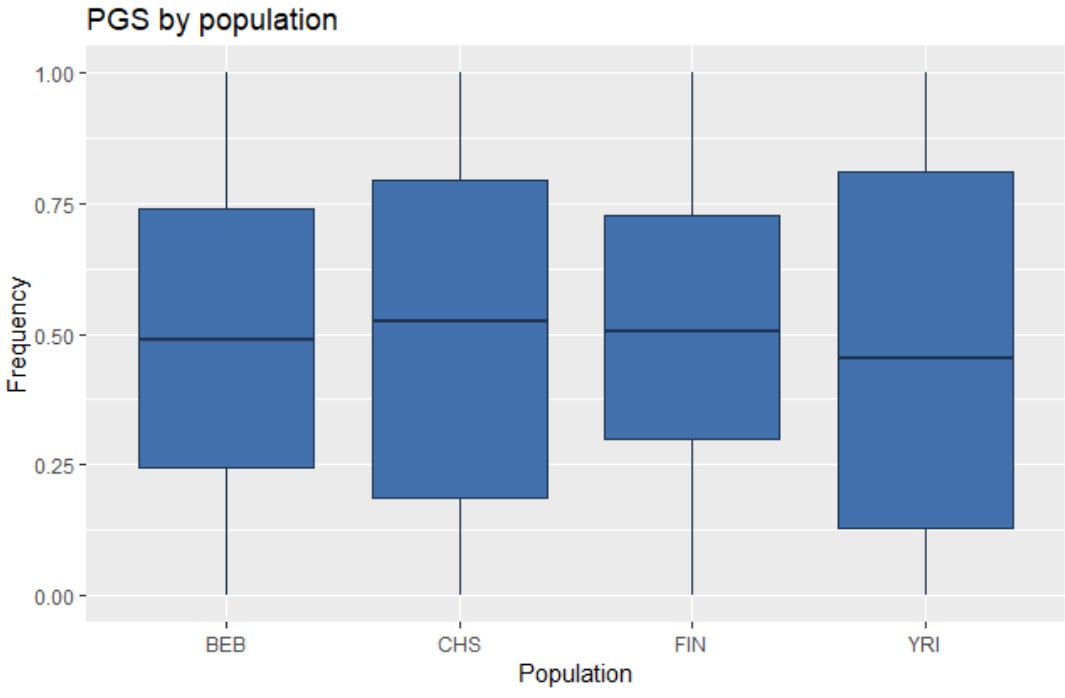

**Figure 6.** Average frequencies by population.

*3.4. GWAS Significance and r x IQ*

The relationship between GWAS significance and predictive power was explored using a meta-correlational approach. In order to maximize variance in p values, the full set of GWAS lead SNPs (N = 9552) was used. Polygenic scores were calculated for 39 significance quantiles (p = 0.025 to 0.975 in

0.025 steps). The correlation coefficients between each score and the population IQ were computed. In turn, the correlation between the correlation coefficient and the quantile was computed, yielding a weak but significant correlation (Spearman's r = −0.34, p = 0.036) (Figure 7).

The mean correlation coefficient (between each set and IQ) was r = 0.470, whereas the correlation coefficient of the full set was r = 0.868.

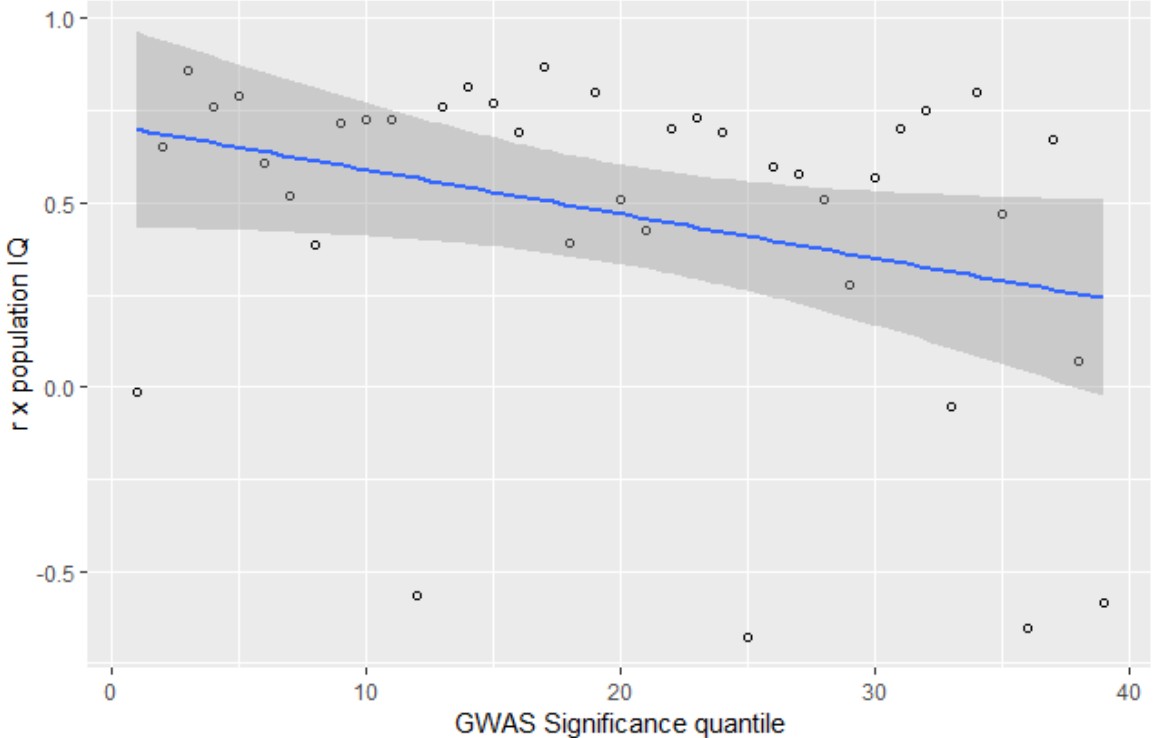

**Figure 7.** Meta-correlation between significance and correlation coefficients.

Height

The correlation between Height PGS and EDU3 was r = −0.288 (Figure.8). The Beta (−0.459) in the regression controlling for SAC (F = 41.82, p-value: < 2.2 × $10^{-16}$) was significant (p < 1.17 × $10^{-7}$), despite large SAC (Fst Beta = 0.738).

The correlation between Height PGS and average phenotypic height was r = 0.486.

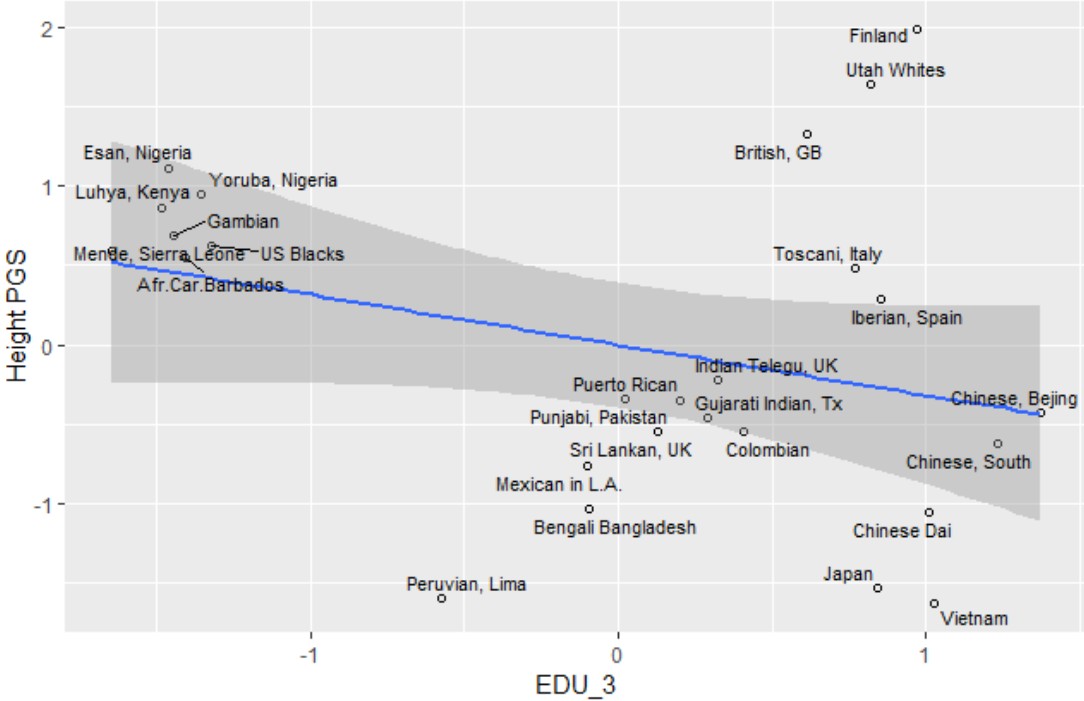

**Figure 8.** Correlation between EDU3 and Height PGS.

*3.5. Socioeconomic Factors*

Nested models were run with a reduced model having EDU PGS as predictor and full models having 1) EDU PGS + Human Development Index; 2) EDU PGS + Total protein consumption; and 3) EDU PGS + Child Mortality. The same was done for height. Since there was a high degree of multicollinearity, only one extra predictor variable was used in each model.

A chi-square test ("anova" function in R) was used to test whether the reduction in the residual sum of squares was statistically significant (as shown by the p value, Tables 3 and 4, last column).

**Table 3.** Nested regression models (EDU).

| EDU PGS | HDI | Total Protein | Child Mortality | R^2 | Sum of Squares |
|---|---|---|---|---|---|
| 0.661*** | 0.371** | | | 0.865 | 473.55 |
| 0.889*** | | | | 0.782 | 807.13 |
| | | | | | P = 0.0012 |
| 0.717*** | | 0.354*** | | 0.885 | 399.97 |
| 0.896*** | | | | 0.793 | 759.19 |
| | | | | | $P = 3.6 \times 10^{-5}$ |
| 0.702*** | | | −0.279* | 0.842 | 640.12 |
| 0.906*** | | | | 0.812 | 509.24 |
| | | | | | P = 0.031 |

Signif. codes: 0 '***' 0.001 '**' 0.01 '*' 0.05 '.' 0.1 ' ' 1.

**Table 4.** Nested regression models (Height).

| Height PGS | HDI | Total Protein | Child Mortality | R^2 | Sum of Squares |
|---|---|---|---|---|---|
| 0.486*** | 0.711*** | | | 0.849 | 85.62 |
| 0.610** | | | | 0.343 | 390.35 |
| | | | | | $P = 5.07 \times 10^{-8}$ |
| 0.238** | | 0.819** | | 0.896 | 58.36 |
| 0.614** | | | | 0.346 | 386.54 |
| | | | | | $P = 3.08 \times 10^{-9}$ |

| | | | |
|---|---|---|---|
| 0.627*** | −0.703*** | 0.837 | 366.77 |
| 0.599** | | 0.327 | 84.29 |
| | | | P = 1.74 × 10⁻⁷ |

Gnomad

Among the 2416 GWAS significant SNPs (Lee et al., 2018) [7] 2404 SNPs were found in the gnomAD dataset. The polygenic scores are reported in Table 5. Since this is a novel dataset, population IQ estimates were generated according to the latest evidence. The correlation between PGS (GWAS sig.) and population IQ is r = 0.979 (N = 8).

**T**able 5 reports the PGS for the eight gnomAD populations.

| Population | IQ | PGS (GWAS sig.) | PGS Clumped |
|---|---|---|---|
| Finnish | 102 (Dutton and Kirkegaard, 2014)[33] | 49.456 | 50.315 |
| Ashkenazi | 110 (Dunkel et al, 2019)[34] | 50.038 | 50.805 |
| Southern Europe | 97(Lynn & Vanhanen, 2012)[35] | 49.119 | 50.056 |
| Estonia | 101(Becker, 2019)[36] | 49.248 | 50.14 |
| NW European | 100(Dutton and Kirkegaard, 2014)[33] | 49.215 | 50.097 |
| African (American) | 85 | 47.414 | 47.656 |
| Latino | 93 (Richwine, 2009)[37] | 48.654 | 49.294 |
| East Asian | 105 (Lynn & Vanhanen, 2012)[35] | 49.750 | 50.076 |

The correlation with population IQ is shown in figure 9.

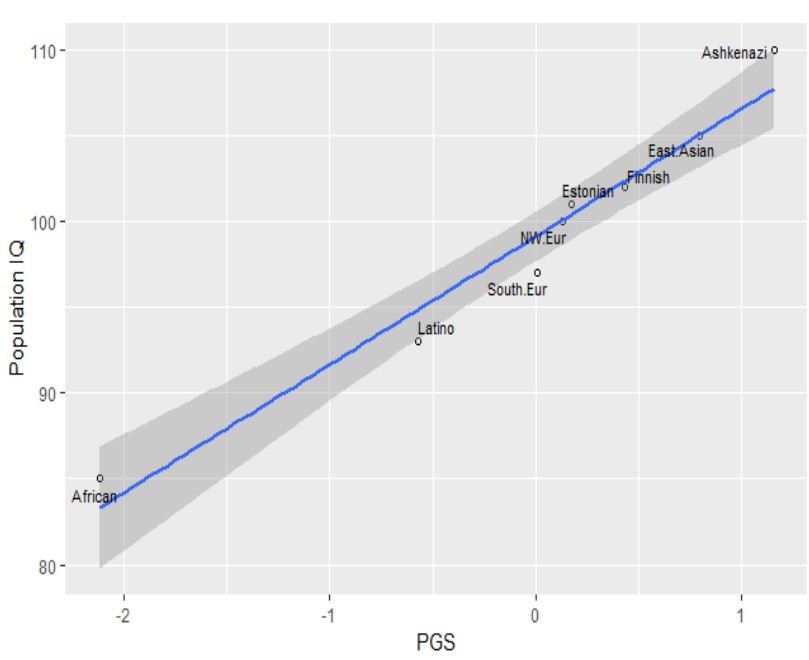

**Figure 9.** Correlation between EDU PGS (gnomAD) and population IQ.

One-way ANOVA was carried out after removing the closely related populations from the European cluster and the heavily admixed Latino population, thus keeping 4 populations: NW European; Ashkenazi Jews; African (American); East Asian. The subset of LD clumped SNPs (N = 1267) reported in Lee et al. (2018) [7] were used to minimize dependence between data points that is present in the full set of GWAS significant lead SNPs. A significant model emerged (F = 2.642; P = 0.0477).

*3.6.GWAS Significance and r x IQ (gnomAD)*

Polygenic scores were calculated for 19 significance quantiles (0.05 to 0.95 in 0.05 steps). The correlation coefficients between each score and the population IQ were computed. In turn, the correlation between the correlation coefficient and the quantile was computed, yielding a weak (but not significant) negative correlation (Spearman's r = −0.33, p = 0.173) (Figure 10). It is evident that the subset's predictive power is lower than that of the full set even at the highest quantiles.

The mean correlation coefficient (between each set and IQ) was r = 0.601, whereas the correlation coefficient of the full set was r = 0.931.

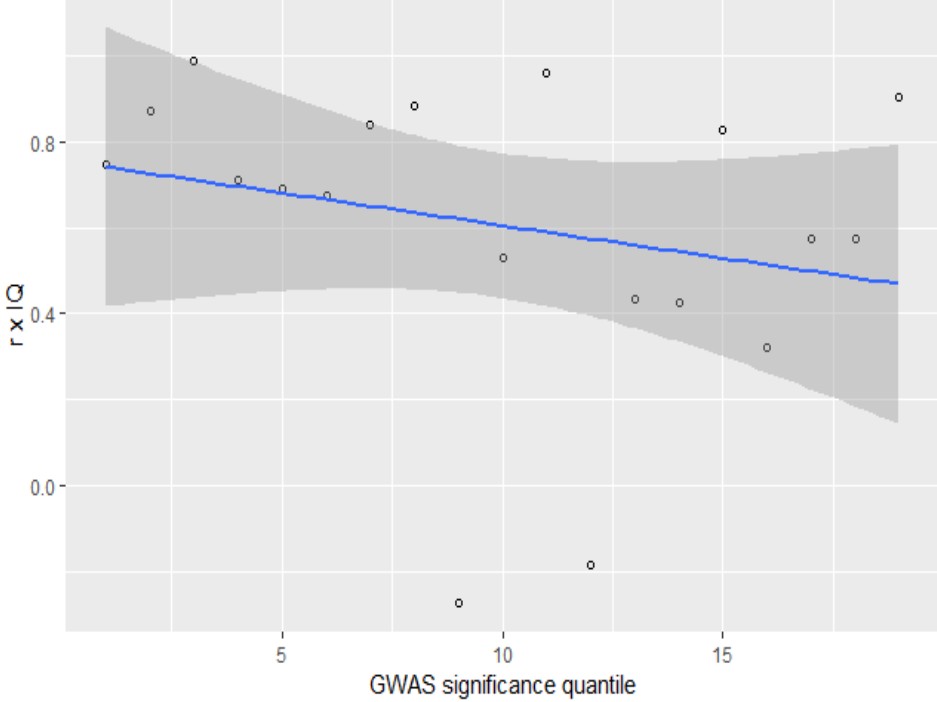

**Figure 10.** Meta-correlation between significance and correlation coefficients.

HGDP-CEPH

The 2411 SNPs (Lee et al., 2018) [7] were searched on the HGDP-CEPH database using the SPSmart browser. In total, 127 SNPs were retrieved and polygenic scores were computed for single populations and continental clusters (Figures.11 and 12). As a check of the robustness of the PGS calculation, a PGS for human height using the latest GWAS (Wood et al., 2014) [38] was computed and it was compared to the PGS calculated by Berg and Coop (2014) [10] for height. These were found to be highly similar (r = 0.84). A significant amount of SAC was detected (Mantel's r = 0.372, p = 0.001).

Positive correlations were found with latitude (r = 0.57) and a weak negative correlation with distance from East Africa (r = −0.29). As no reliable estimates of IQ were found for most of the 52 populations, no attempt was made to include this phenotype in the analysis of the HGDP sample.

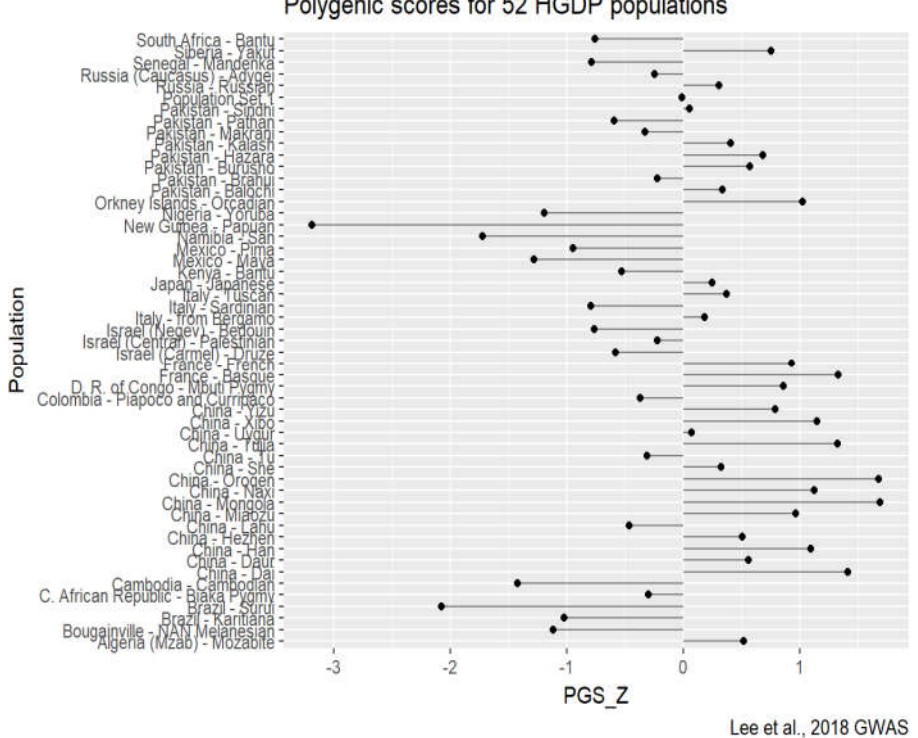

**Figure 11.** Polygenic scores for HGDP (ALFRED) populations.

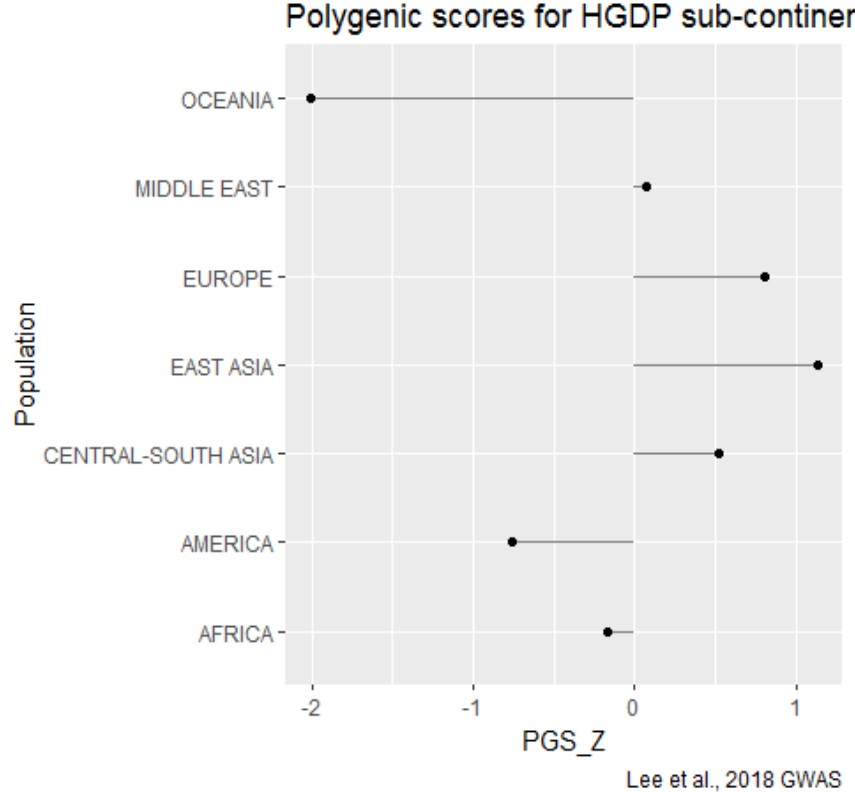

**Figure 12.** Polygenic scores by continent.

## 4. Discussion

The calculation of population-level polygenic scores (average allele frequencies with positive GWAS beta) is a promising and quick approach to test signals of polygenic adaptation. The results clearly showed population differences in PGS (Figure 3), which correlated with estimates of average population IQ (Figure 2) and students performance on standardized tests of mathematics, reading and science (r = 0.9 and 0.8, respectively).

The EDU3 polygenic score was the most robust to tests of spatial autocorrelation (Table 2), that is, it predicted population IQ also after removing SAC by partial Mantel test via Fst distances [9,28]. In fact, when IQ was regressed on the PGS and Fst distances, the latter lost all the predictive value (Beta = 0.045, p = 0.539), whereas the former retained high predictive power (Beta = 0.68, p = <2.2e × $10^{-16}$). Additionally, partial and semi-partial correlation were also used as an alternative method to remove SAC, and both yielded significant correlation coefficients (respectively, r= 0.506; p = 7.85 × $10^{-18}$ and r = 0.409; p = 1.304 × $10^{-11}$). Similar results were obtained by using Learning index instead of IQ.

The polygenic score computed for MTAG-derived Cognitive Performance SNPs was highly correlated to EDU3 (r = 0.92) and to population IQ (r = 0.86).

More strength is given to the present findings by the high replicability of the same polygenic model from a previous publication (Piffer, 2015 [9]). Piffer (2015) [9] calculated polygenic and factor scores for 1000 Genomes population using the data available at the time from GWAS that relied on much smaller sample size. Since the correlation between the present study's EDU3 and previous estimates of polygenic scores score are very high (r = 0.95–0.98), and the correlations with the average population IQ estimates from the previous study are similarly high (Figure 2), the claim that this finding could be post-hoc is ruled out.

A Monte Carlo simulation using 943 PGS computed from sets of 2411 randomly matched SNP showed that none of them reached a correlation with population IQ equal to or higher than EDU's (r = 0.886), corresponding to p = 0.001.

Height was used as a control variable, due to its similar polygenic nature to cognitive ability. The height PGS was positively correlated with average population height but it had a weak negative association with EDU PGS and with population IQ (Figure. 8). The lack of a relationship between the two PGS and the lack of an association with a phenotype (i.e. IQ) different from the GWAS trait (i.e. height) suggests that potential biases (e.g. favoring the European reference population) in the GWAS procedure do not drive the correlation between EDU PGS and IQ. Remarkably, whilst East Asians scored at the top of the EDU PGS, they also scored at the bottom of the height PGS, in accordance with their relatively smaller physical size.

The results were replicated in a new dataset (gnomAD), comprising a much larger sample of individuals, where the correlation between population IQ and PGS was 0.98 (Table 5 and Figure 9). This dataset included a sample of 145 Ashkenazi Jewish individuals. The IQ of Ashkenazi Jews has been estimated to be around 110 [34]. Remarkably, their EDU polygenic score was the highest in our sample, corresponding to a predicted score of about 108, mirroring preliminary results from a smaller (N = 53) sample (Dunkel et al., 2019) [34].

The large sample of Finnish individuals present in gnomAD also replicated their polygenic score advantage found in 1000 Genomes, closely mirroring the advantage over other European populations observed in scholastic aptitude and intelligence tests [33].

One-way ANOVA found differences in mean allele frequencies between populations both in 1000 Genomes and gnomAD dataset.

The overall results were replicated using a larger sample of populations from the HGDP-CEPH panel (N = 52), which showed similar population and continental rankings of polygenic scores (Figures 11 and 12). A positive correlation with latitude was found (r = 0.57), but a small negative one with distance from Eastern Africa (r = −0.29). The latter finding casts doubt on the hypothesis that migratory patterns account for population differences in polygenic scores. The positive correlation with latitude could reflect selection pressures due to climate, but further evaluating this hypothesis

would require exhaustive simulations with null SNPs, using an approach similar to Berg and Coop (2014) [10], which are beyond the scope of this paper.

The much higher East Asian scores (Figure 3 and Table 5) suggests that strong selection pressure on East Asians continued after the East Asian-Native American split, about 15 kya at the earliest (the earliest estimate of a migration across the Bering strait into the Americas) [8]). This date has been disputed. There is evidence from ancient genomes that the split between northeast Asians and Native Americans happened 20kya when both were still residing in NE Asia, and there was continued gene flow until the Native American group crossed the Bering strait into North America around 15-14 kya [39]. It is possible that the extremely low population density in the Americas reduced intraspecific competition (hence, selection pressure on cognitive ability was lower), but this topic is widely open to debate and speculation.

A limitation of the present study is its reliance on estimates of population IQ as the phenotypic variable, which are not perfectly accurate, besides potentially reflecting environmental and economic differences between populations. Moreover, the EA GWAS can capture genetic variation that contributes to educational attainment via mechanisms other than IQ (i.e. conscientiousness).

The moderate (11–13%) amount of variance explained at the individual level by the full set and the subset of significant hits (3.2%) (Lee et al., 2018) [7] is not an issue of primary importance. Indeed, predicting group-level variance is different from predicting variance within a group. The important difference between trans-ethnic and within population phenotypic predictions—as measured by the amount of variance explained—is that while the latter is maximized by using the full set of GWAS SNPs, in the former, using low significance SNPs risks introducing too much noise, such as elements of drift or migrations, that would dilute the signal derived from selection pressures on a specific trait. Hence, the two problems require different approaches to SNP selection and optimal PGS construction.

Deciding the optimum number of SNPs can be done empirically (by picking the significance threshold that results in the highest correlation with the average phenotypic population value). In our data, there was degradation of signal across significance quantiles, as shown by a weak trend for lower significance SNPs to have lower correlation with population IQ (Figures 7 and 10). More remarkably, the PGS generated from most SNP subsets had lower predictive power than that of the full set. For example, using the full set of 10k lead SNPs in 1000 Genomes, the average correlation coefficient (between each set's PGS and IQ) was $r = 0.470$, whereas the correlation coefficient of the full set with IQ was $r = 0.868$. In the gnomAD dataset, 0.61 and the full (clumped) PGS's correlation was 0.93.

This suggests the presence of a trade-off between quantity and quality of SNPs relative to gains in predictive power. That is, a larger number of SNPs increases the signal, yet it introduces more noise due to inclusion of lower significance SNPs. The maximal predictive power was reached by selecting SNPs which met the conventional GWAS significance threshold ($P < 5 \times 10^{-8}$), whilst picking higher significance SNPs reduced the predictive power (due to reduced number of SNPs). Although traditionally weighing by effect size is the most commonly employed weighting method, and since no increase in predictive accuracy was observed in this study for the effect size weighted PGS vs. the unweighted (raw frequency) PGS, p value weighting should be used as a valid alternative for PGS computation, particularly when including SNPs that are below the conventional GWAS significance threshold. The predictive accuracy of the PGS in this study is saturated by the high correlation with population IQ, but methods could be used in other studies to improve PGS construction. Reviewer 2 suggested the following procedure for optimal PGS construction that could be used in future studies: "Start with the quantile that has the most significant SNPs, and then add quantiles in declining order of genome-wide significance. Initially, adding quantiles will improve prediction, but after a certain point, adding more quantiles will make prediction worse. At that inflection point you have the optimal PGS".

A persistent issue is that the trans-ethnic validity of PGS is compromised by LD decay. This is the decay in linkage disequilibrium with time, meaning that the older the causal polymorphism, the lower the level of linkage disequilibrium due to recombination events that occur with constant

probability in every generation. As a consequence, linkage patterns can be different in different populations, especially those that separated a long time ago and that underwent population bottlenecks after separation.

The effect of LD decay on comparison of risk alleles between populations is still unclear. Since most GWAS hits are actually tag SNPs, decay in LD implies that the causal SNPs will be less efficiently flagged by the tag SNPs until the tag SNPs will resemble a sample of random SNPs. With less significant associations, it is not only more likely that the distance between the GWAS hit and the causal polymorphism is larger and linkage is weaker in the European-origin populations that are represented in the GWASs, but it is also more likely that the linkage phase is different in different races.

LD is sensitive to coverage and in older studies using low coverage genomic data (e.g. 1000 Genomes phase 1), it was found to reduce the reproducibility of findings [40]. However, contemporary GWAS use higher coverage data (e.g. 1000 Genomes phase 3); hence, this issue is less important.

Moreover, simulations found that the effect of LD decay on true causal variants was null to negligible [41]. The present study, by focusing only on the most significant hits (N = 2411), increased the likelihood of hitting on or very close to causal variants, hence reducing the artifact of LD decay. Moreover, the analysis was replicated using a set of putatively causal SNPs (N = 125) from the Lee et al.(2018) [7] paper. The correlation with population IQ was still high (r = 0.82 and 0.85 with the weighted and unweighted PGS, respectively) although not as high as that achieved by the larger set: this could be caused by the loss of signal due to the much smaller number of SNPs.

In contrast, Lee et al. (2018) [7] analyzed the association between EDU PGS and years of education in an older African American sample. Given their use of all SNPs regardless of significance, it is not surprising that the cross-ethnic validity of their scores was drastically reduced [7]. Moreover, since the heritability of education among older African Americans is unknown and the predictive validity of PGS scores depend on the population specific trait heritability, we do not know if the reduction in trans-ethnic validity was due to a reduced heritability.

In fact, it is well known that polygenic scores perform better in European populations, and prediction accuracy is reduced by approximately 2 to 5 fold in East Asian and African American populations, respectively [42].

A recent study has replicated the validity of Lee et al. (2018) [7]'s PGS on a sample of African Americans; the authors found that a higher EA PGS was associated with higher probability of college completion and math performance, although not with reading achievement. There was also a negative association with criminal record status. However, there was an attenuation compared to the PGS effect among White subjects. The authors attribute this to various potential factors besides LD decay: 1) The study used a sample low in socioeconomic status (SES), where shared environment plays a bigger role than in high SES environments [43]; and 2) Measures of math and reading performance were obtained in early childhood, a developmental period during which the importance of genetic influences on intelligence is lower (and that of shared environment is higher) compared to young adulthood [43]. Nonetheless, this study provides evidence for the (partial) transferability of EA polygenic scores to African Americans.

Additionally, trans-ethnic GWAS meta-analyses on other traits have also found genetic variants with little heterogeneity between ancestry groups [44,45]. A recent GWAS of schizophrenia found that approximately 95% of SNPs from a Western GWAS had consistent direction of effect in the Chinese sample, and this was significant for about half of those (Li et al., 2017) [46].

We postulate that this common core of causal genetic variants with trans-ethnically homogeneous effects is what drives the association between average trait values and population IQ or height, and the group differences in mean PGS. This would be superimposed on a background of heterogeneity of allelic effects, thus adding noise to the data.

In fact, LD decay is expected (from a theoretical perspective) to create noise and follow drift and not to produce a bias necessarily in a direction that favours the hypothesis of this study [40]. Since the frequency of the average SNP allele is 50%, the tag SNPs will tend to converge towards an average

frequency of 50%, with increasing LD decay. The implication of this for our analysis is that when the polygenic scores are below 50%, our estimates will be inflated, and vice-versa for the polygenic scores which are above 50%, because LD decay pushes the polygenic scores up (or down) towards the background frequency of 0.5. In the present case, the average frequency of alleles with positive effect is around 50% (49.7%) for CEU and CHS (49.7% and 50.1%); hence, LD decay should produce only a tiny bias in the estimate (upward and downward bias, respectively). However, the allele frequencies of causal SNPs in other (non-European) populations that are farther away from 50% will produce a stronger bias. For example, the average YRI frequency is 47.5%, so LD decay produces an overestimate of the PGS. In other words, the frequency of causal SNPs is expected to be lower than 47.5%.

A preliminary test of this hypothesis was carried out by employing 125 candidate causal SNPs from the Lee et al. (2018) [7] GWAS. The unweighted EDU3-Causal PGS difference was negatively correlated ($r = -0.62$) to the unweighted EDU3 PGS. In other words, the lower the frequency of the EDU3 PGS, the higher the estimate in the larger set than in the set of putatively causal SNPs, suggesting that LD decay leads to an overestimation of the PGS in the populations with lower PGS, as predicted by theory.

Adding socioeconomic variables to the model slightly (but significantly) increased the predictive power (from 78–80% to 85–89%), although the PGS explained twice as much of the variance (70% vs. 35%) as those of HDI, average protein consumption or child mortality (Table 3). The reverse was the case for height, where socioeconomic factors explained much more of the variance in average height than the PGS (Table 4). This is in line with heritability studies which show the importance of the shared environment for adult height even in rich Western countries (about 10%) and more so in non-Western countries [47], and the dramatic secular trend in height. On the other hand, the shared environmental impact on IQ or g in adulthood is typically found to be near zero [48], although this might not be the case in developing countries or deprived rearing environments. Indeed, the IQ of African Americans appears to be higher than what is predicted by the PGS (Figure 2), which suggests this cannot be explained by European admixture alone, but it could be the result of enjoying better nutrition or education infrastructure compared to native Africans. Another explanation is heterosis ("hybrid vigor"), that is the increase in fitness observed in hybrid offspring thanks to the reduced expression of homozygous deleterious recessive alleles [49]. The Sri Lankan UK population also constitute an outlier, because their IQ is lower than that predicted by the PGS. This does not contradict the previous statement, because the IQ estimate obtained from Piffer (2015) [9] was based on native Sri Lankans, since estimates for Sri Lankans living in the UK were not available. Given the moderate impact of environment, it is likely that the IQ of Sri Lankans living in the UK is actually higher than that of native Sri Lankans.

Testing more sophisticated models with larger sets of socioeconomic variables would go beyond the scope of this paper but it is an interesting direction for future research.

Future GWAS studies should be carried out on non-European populations. Indeed, trans-ethnic GWASs are a promising resource for the identification of alleles with homogeneous and heterogeneous effects and the computation of population-specific polygenic scores. Specifically, they would enable us to include SNPs that are polymorphic only in some populations, and to find the causal SNPs that have the same causal effect in all populations.

**Supplementary Materials:** https://osf.io/g9yx8/

**Funding:** This research received no external funding.

**Acknowledgments:** I thank John Fuerst for copyediting the manuscript and providing suggestions that greatly increased its clarity.

**Conflicts of Interest:** The authors declare no conflict of interest

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
