# Peer review of "Evidence for Recent Polygenic Selection on Educational Attainment and Intelligence Inferred from Gwas Hits: A Replication of Previous Findings Using Recent Data"

_psych, doi:10.3390/psych1010005_

Round 1
Reviewer 1 Report
This manuscript is a replication/extension of previous studies by the author using additional data sets and similar approaches, with stronger conclusions. The datasets are extensive and the methodologies are sound and include appropriate controls and analyses to address potential caveats, all of which justify the conclusions. Overall, this is a worthwhile addition to the study of the genetic basis of intelligence especially as it relates to group differences. I would suggest that authors address the following relatively minor issues in their final revised manuscript:
The outliers in Figure 2 may be worth some discussion. For example, US Blacks have higher IQ than predicted by EDU3 PGS, especially relative to African Blacks. This could be just noise but could it also suggest that good educational infrastructure in the US has helped to improve their IQ relative to African blacks? This would add to the argument of both genes and environment contributing to IQ outcome. Sri Lankans in UK actually have lower IQ than what their EDU3 PGS would predict, which seems to run counter to the idea of good education infrastructure raising IQ. But then, all of these outliers could just be noise.
I noticed that there are no Jews in Figure 2 and other similar figures/tables. I wonder if this is because the genetic data for Jews are not available or whether there is some other reason for this. The author should comment on why such an interesting group is missing from their main figures.
EDU_3 appears a few times in some figures. I presume it is the same as EDU3 which the author clearly defined in the manuscript. If so, EDU_3 should be changed to EDU3.
Author Response
Point 1: The outliers in Figure 2 may be worth some discussion. For example, US Blacks have higher IQ than predicted by EDU3 PGS, especially relative to African Blacks. This could be just noise but could it also suggest that good educational infrastructure in the US has helped to improve their IQ relative to African blacks? This would add to the argument of both genes and environment contributing to IQ outcome. Sri Lankans in UK actually have lower IQ than what their EDU3 PGS would predict, which seems to run counter to the idea of good education infrastructure raising IQ. But then, all of these outliers could just be noise.
Reply:This is a good observation. Indeed, US Blacks are on average 20% admixed with European Americans. But their IQ is higher than what is predicted even by their PGS. This could be due to hybrid vigor or to good education infrastructure raising IQ. The low Sri Lankan IQ needs some explanation.The1000 Genomes dataset includes a sample of Sri Lankan Tamil living in the UK. However, lacking IQ statistics for Sri Lankan living in the UK, I used the IQ reported for native Sri Lankans living in Sri Lanka because the individuals are not admixed with local (British) population. However, I acknowledge that this is a limitation because (thanks to a phenomenon called selective migration), immigrants are not necesIsarily representative of the native Sri Lankan population. It is likely that the IQ of Sri Lankans living in the UK is higher.
I added the following sentence to the discussion: “ Indeed, the IQ of African Americans appears to be higher than what is predicted by the PGS (figure 2), which suggests this cannot be explained by European admixture alone but it could be the result of enjoying better nutrition or education infrastructure compared to native Africans. Another explanation is heterosis ("hybrid vigor"), that is the increase in fitness observed in hybrid offspring thanks to reduced the expression of homozygous deleterious recessive alleles (Shull, 1948). The Sri Lankan UK population also constitute an outlier, because their IQ is lower than what is predicted by the PGS. This does not contradict the previous statement because the IQ estimate obtained from Piffer (2015) was based on native Sri Lankans, since estimates for Sri Lankans living in the UK were not available. Given the moderate impact of environment, it is likely that the IQ of Sri Lankans living in the UK is actually higher than that of native Sri Lankans.”
Point 2: I noticed that there are no Jews in Figure 2 and other similar figures/tables. I wonder if this is because the genetic data for Jews are not available or whether there is some other reason for this. The author should comment on why such an interesting group is missing from their main figures.
Reply:Figure 2 represents only data from 1000 Genomes and this dataset does not include Jews. The Ashkenazi Jewish sample is included in the gnomAD data, which is shown in table 5 and in figure 9.
Point 3: EDU_3 appears a few times in some figures. I presume it is the same as EDU3 which the author clearly defined in the manuscript. If so, EDU_3 should be changed to EDU3.
Reply:Done

Reviewer 2 Report
This is an important paper from the cutting edge of molecular genetics. It uses the latest GWAS results to confirm results that had been obtained with earlier data. The paper needs some brushing up. In particular, it is very technical and in its present form is not optimal to make this field accessible to people who are not acquainted with molecular genetics. There are many changes big and small that can and should be made to make the subject more accessible and put it into a broader perspective.
And now the specifics:
Lines 12-14: Not clear. Does “9 SNPs within genomic regions” mean 9 different genomic regions? In same sentence, does the N=2411 refer to the number of subjects in the GWAS, or the number of SNPs contributing to the polygenic score?
Line 20: In both data set(s)? Do you mean both polygenic scores, or gnomAD and 1000Genomes?
Line 38: “…a decent amount of variance” Any more precise way of expressing this, for example “more than 5% to 10% of the variance”?
Introduction general: Most readers of your article are not molecular geneticists and have only a foggy idea of how genetics works. You should make an effort to make things intelligible for these non-geneticists. For example, you should mention that the GWAS hits are SNPs that in most cases are not causal but are merely linked genetically with causal variants that happen to be located close to the SNP. Most non-experts are not aware of this. More specifically, you should mention some of the limitations of polygenic scores. One is that the subjects in the association studies were overwhelmingly of European descent. This is important for comparisons between countries because some of the non-European genetic variance is missed. Also, the linkage phase can be different in different races.
Line 91: Give website from which these files were downloaded.
Line 111: How do you get more than 2 million “unlinked” SNPs?
Line 123: In this method, do you use the average Fst values for a large number of SNPs?
Line 124: “PGS distances were employed as the index of genetic distance”. Here, better add that this is genetic distance for the education-relevant polymorphisms, while the Fst (averaged across many loci) is a measure of genome-wide genetic distance, if that’s what it means. The point, as I understand it, is to compare these two kinds of genetic distance, with mismatches indicating selection acting on education-related genetic variation. Make sure readers get that point.
Figure 1: This would look better as a table, unless you can get better quality of reproduction for this figure.
Lines 149-156: A lot of acronyms are used here that force the reader to read back at least 1 or 2 sentences. Try to make this easier to read either by spelling out things (e.g., “cognitive performance” instead of CP) or, at least, by putting (CP) after first mention of cognitive performance.
Figure 3: Indicate in the figure legend that this is the unweighted polygenic score.
Lines 162-169: Here you describe the correlations of the different polygenic scores with each other and with national IQ. This is indeed important, but somewhere in your paper you should place this into a broader perspective by mentioning how many of the 2411 SNPs have highest frequency and how many have the lowest frequency in Europe, Africa, East Asia and South Asia. And you should report how many of the individual SNPs have positive correlations with national IQ, and how many have negative correlations (and perhaps, how many are virtually uncorrelated with IQ). The same for the restricted set of SNPs that Lee et al. consider good candidates for being causal. This is to give a crude indication of how much variation there is in the species, and how important differences between populations are relative to total genetic variance. For example, if 80% of SNPs have positive and 20% have negative correlations with national IQ, this would indicate that there are very substantial race differences compared to total variation. Things would look different if, for example, 55% of the SNPs have positive correlations and 45% have negative correlations. Such an observation would show that there is huge total variation in the species but very little of this is related to race. Many non-geneticists don’t seem to realize what a huge amount of genetic variation there is in our species, both common polymorphisms with small effects and rare variants with sometimes large effects. If there is an opportunity to educate your readers about this, make use of it!
Table 2: You should state in the table heading what the dependent variable is in these regressions and the number of cases, in addition to what has been controlled. This table seems to show 3 regression models. Make horizontal lines to separate them, otherwise readers get confused.
Line 205: Acronyms! What is BEB, for example? Better spell out the four population names.
Line 215: Are these genome-wide significance values for the SNPs?
Line 218: To make it easier for the reader to follow, insert one or two sentences stating that this amounts to a weak tendency for SNPs with better statistical significance to be better predictors of IQ. Also, in this exercise you divide up the SNPs according to statistical significance brackets, which means you have somewhat different numbers of SNPs in each bracket. It would be better to rank the SNPs and then define the brackets with equal numbers of SNPs in each. This is because everything else being equal, a polygenic score computed from a smaller number of SNPs will be less accurate than one from a larger number of SNPs and therefore will produce lower correlations with the dependent variable (national IQ). Same for the procedure described in line 259-263.
Lines 224-227: Sentence line 227 should come first because this is the most fundamental piece of information about the height PGS. In the second sentence you mention the results of a regression. Is this average height in the country predicted with Fst and polygenic score? What is the number of cases in this regression? Such info is important to help the reader figure out what exactly you did. You mention a slightly negative correlation between height and Edu polygenic scores. This is a bit surprising because phenotypically at the individual level there is a slight positive correlation between height and Edu/IQ. A case of Simpson’s paradox. You may want to discuss this finding somewhere. Is anything known about genetic overlap between height and Edu/IQ? This might mediate some of the phenotypic correlation at the individual level, for example if some SNPs influence both general growth and brain size.
Line 342-349: One specific reason for avoiding the less significant GWAS hits in population-level studies is that the linkage phase may be different in different races, a problem that plays no role when comparing individuals within countries. With weak associations, it is not only more likely that the distance between the GWAS hit and the causal polymorphism is larger and linkage is weaker in the European-origin populations that are represented in the GWASs, but it is also more likely that the linkage phase is different in different races.
Line 350-358: You can propose the following more formal procedure for constructing the optimal PGS: Start with the quantile that has the most significant SNPs, and then add quantiles in declining order of genome-wide significance. See whether adding quantiles improves or worsens prediction. Initially, adding quantiles will improve prediction, but after a certain point, adding more quantiles will make prediction worse. At that inflection point you have the optimal PGS.
Line 368-372: Linkage decay by itself will only reduce prediction, and will do so to the same extent in all populations that are descended from the original mutant (assuming the causal polymorphism is younger than the ones with which it is linked). What messes things up is that there were serial population bottlenecks where a rare recombinant may have been the only survivor carrying the derived allele. In that case the descendants of this bottlenecked population will no longer have the original linkage phase. For example, if ancestral Europeans but not Africans went through such a bottleneck, the linkage phase may be opposite in Europe and Africa, and what predicts higher IQ in Europe predicts lower IQ in Africa. Therefore the urgent need for GWASs in non-European populations, both to find the causal polymorphisms (assumed to predict in the same direction everywhere) and to improve genomic prediction for non-Europeans. Eventually, we want “color-blind” polygenic scores that are computed from causal polymorphisms including those that are polymorphic in only some human populations.
This is an important paper from the cutting edge of molecular genetics. It uses the latest GWAS results to confirm results that had been obtained with earlier data. The paper needs some brushing up. In particular, it is very technical and in its present form is not optimal to make this field accessible to people who are not acquainted with molecular genetics. There are many changes big and small that can and should be made to make the subject more accessible and put it into a broader perspective.
And now the specifics:
Lines 12-14: Not clear. Does “9 SNPs within genomic regions” mean 9 different genomic regions? In same sentence, does the N=2411 refer to the number of subjects in the GWAS, or the number of SNPs contributing to the polygenic score?
Line 20: In both data set(s)? Do you mean both polygenic scores, or gnomAD and 1000Genomes?
Line 38: “…a decent amount of variance” Any more precise way of expressing this, for example “more than 5% to 10% of the variance”?
Introduction general: Most readers of your article are not molecular geneticists and have only a foggy idea of how genetics works. You should make an effort to make things intelligible for these non-geneticists. For example, you should mention that the GWAS hits are SNPs that in most cases are not causal but are merely linked genetically with causal variants that happen to be located close to the SNP. Most non-experts are not aware of this. More specifically, you should mention some of the limitations of polygenic scores. One is that the subjects in the association studies were overwhelmingly of European descent. This is important for comparisons between countries because some of the non-European genetic variance is missed. Also, the linkage phase can be different in different races.
Line 91: Give website from which these files were downloaded.
Line 111: How do you get more than 2 million “unlinked” SNPs?
Line 123: In this method, do you use the average Fst values for a large number of SNPs?
Line 124: “PGS distances were employed as the index of genetic distance”. Here, better add that this is genetic distance for the education-relevant polymorphisms, while the Fst (averaged across many loci) is a measure of genome-wide genetic distance, if that’s what it means. The point, as I understand it, is to compare these two kinds of genetic distance, with mismatches indicating selection acting on education-related genetic variation. Make sure readers get that point.
Figure 1: This would look better as a table, unless you can get better quality of reproduction for this figure.
Lines 149-156: A lot of acronyms are used here that force the reader to read back at least 1 or 2 sentences. Try to make this easier to read either by spelling out things (e.g., “cognitive performance” instead of CP) or, at least, by putting (CP) after first mention of cognitive performance.
Figure 3: Indicate in the figure legend that this is the unweighted polygenic score.
Lines 162-169: Here you describe the correlations of the different polygenic scores with each other and with national IQ. This is indeed important, but somewhere in your paper you should place this into a broader perspective by mentioning how many of the 2411 SNPs have highest frequency and how many have the lowest frequency in Europe, Africa, East Asia and South Asia. And you should report how many of the individual SNPs have positive correlations with national IQ, and how many have negative correlations (and perhaps, how many are virtually uncorrelated with IQ). The same for the restricted set of SNPs that Lee et al. consider good candidates for being causal. This is to give a crude indication of how much variation there is in the species, and how important differences between populations are relative to total genetic variance. For example, if 80% of SNPs have positive and 20% have negative correlations with national IQ, this would indicate that there are very substantial race differences compared to total variation. Things would look different if, for example, 55% of the SNPs have positive correlations and 45% have negative correlations. Such an observation would show that there is huge total variation in the species but very little of this is related to race. Many non-geneticists don’t seem to realize what a huge amount of genetic variation there is in our species, both common polymorphisms with small effects and rare variants with sometimes large effects. If there is an opportunity to educate your readers about this, make use of it!
Table 2: You should state in the table heading what the dependent variable is in these regressions and the number of cases, in addition to what has been controlled. This table seems to show 3 regression models. Make horizontal lines to separate them, otherwise readers get confused.
Line 205: Acronyms! What is BEB, for example? Better spell out the four population names.
Line 215: Are these genome-wide significance values for the SNPs?
Line 218: To make it easier for the reader to follow, insert one or two sentences stating that this amounts to a weak tendency for SNPs with better statistical significance to be better predictors of IQ. Also, in this exercise you divide up the SNPs according to statistical significance brackets, which means you have somewhat different numbers of SNPs in each bracket. It would be better to rank the SNPs and then define the brackets with equal numbers of SNPs in each. This is because everything else being equal, a polygenic score computed from a smaller number of SNPs will be less accurate than one from a larger number of SNPs and therefore will produce lower correlations with the dependent variable (national IQ). Same for the procedure described in line 259-263.
Lines 224-227: Sentence line 227 should come first because this is the most fundamental piece of information about the height PGS. In the second sentence you mention the results of a regression. Is this average height in the country predicted with Fst and polygenic score? What is the number of cases in this regression? Such info is important to help the reader figure out what exactly you did. You mention a slightly negative correlation between height and Edu polygenic scores. This is a bit surprising because phenotypically at the individual level there is a slight positive correlation between height and Edu/IQ. A case of Simpson’s paradox. You may want to discuss this finding somewhere. Is anything known about genetic overlap between height and Edu/IQ? This might mediate some of the phenotypic correlation at the individual level, for example if some SNPs influence both general growth and brain size.
Line 342-349: One specific reason for avoiding the less significant GWAS hits in population-level studies is that the linkage phase may be different in different races, a problem that plays no role when comparing individuals within countries. With weak associations, it is not only more likely that the distance between the GWAS hit and the causal polymorphism is larger and linkage is weaker in the European-origin populations that are represented in the GWASs, but it is also more likely that the linkage phase is different in different races.
Line 350-358: You can propose the following more formal procedure for constructing the optimal PGS: Start with the quantile that has the most significant SNPs, and then add quantiles in declining order of genome-wide significance. See whether adding quantiles improves or worsens prediction. Initially, adding quantiles will improve prediction, but after a certain point, adding more quantiles will make prediction worse. At that inflection point you have the optimal PGS.
Line 368-372: Linkage decay by itself will only reduce prediction, and will do so to the same extent in all populations that are descended from the original mutant (assuming the causal polymorphism is younger than the ones with which it is linked). What messes things up is that there were serial population bottlenecks where a rare recombinant may have been the only survivor carrying the derived allele. In that case the descendants of this bottlenecked population will no longer have the original linkage phase. For example, if ancestral Europeans but not Africans went through such a bottleneck, the linkage phase may be opposite in Europe and Africa, and what predicts higher IQ in Europe predicts lower IQ in Africa. Therefore the urgent need for GWASs in non-European populations, both to find the causal polymorphisms (assumed to predict in the same direction everywhere) and to improve genomic prediction for non-Europeans. Eventually, we want “color-blind” polygenic scores that are computed from causal polymorphisms including those that are polymorphic in only some human populations.
v
Author Response
This is an important paper from the cutting edge of molecular genetics. It uses the latest GWAS results to confirm results that had been obtained with earlier data. The paper needs some brushing up. In particular, it is very technical and in its present form is not optimal to make this field accessible to people who are not acquainted with molecular genetics. There are many changes big and small that can and should be made to make the subject more accessible and put it into a broader perspective.
And now the specifics:
Lines 12-14: Not clear. Does “9 SNPs within genomic regions” mean 9 different genomic regions?
Reply:Yes (Changed text accordingly: henceforth "C.T.A"!).
In same sentence, does the N=2411 refer to the number of subjects in the GWAS, or the number of SNPs contributing to the polygenic score?
Reply:Number of SNPs (C.T.A).
Line 20: In both data set(s)? Do you mean both polygenic scores, or gnomAD and 1000Genomes?
Reply:gnomAD and 1000 Genomes (C.T.A).
Line 38: “…a decent amount of variance” Any more precise way of expressing this, for example “more than 5% to 10% of the variance”?
Reply:Agreed. C.T.A
Introduction general: Most readers of your article are not molecular geneticists and have only a foggy idea of how genetics works. You should make an effort to make things intelligible for these non-geneticists. For example, you should mention that the GWAS hits are SNPs that in most cases are not causal but are merely linked genetically with causal variants that happen to be located close to the SNP. Most non-experts are not aware of this. More specifically, you should mention some of the limitations of polygenic scores. One is that the subjects in the association studies were overwhelmingly of European descent. This is important for comparisons between countries because some of the non-European genetic variance is missed. Also, the linkage phase can be different in different races.
Reply:Added explanations of these issues in the introduction. In particular, I explain how population specific variants (those that are polymorphic only to some populations) are not expected to bias the PGS in favor of the reference (in this case European) group and I use the example of the Peruvian SNP for lower stature.
Line 91: Give website from which these files were downloaded.
Reply:Done
Line 111: How do you get more than 2 million “unlinked” SNPs?
Reply:I used SNPSNAP to download the list of 2+million SNPs matching those of the GWAS. Moved sentences so this is clearer.
Line 123: In this method, do you use the average Fst values for a large number of SNPs?
Reply:I use the average Fst values for all SNPs contained in Chromosome 1, as in Piffer(2015). C.T.A.
Line 124: “PGS distances were employed as the index of genetic distance”. Here, better add that this is genetic distance for the education-relevant polymorphisms, while the Fst (averaged across many loci) is a measure of genome-wide genetic distance, if that’s what it means. The point, as I understand it, is to compare these two kinds of genetic distance, with mismatches indicating selection acting on education-related genetic variation. Make sure readers get that point.
Reply:Ok, I tried to make the sentence clearer.
Figure 1: This would look better as a table, unless you can get better quality of reproduction for this figure.
Reply:I think the figure is OK. Otherwise, please tell me how I can change it to make it look nicer.
Lines 149-156: A lot of acronyms are used here that force the reader to read back at least 1 or 2 sentences. Try to make this easier to read either by spelling out things (e.g., “cognitive performance” instead of CP) or, at least, by putting (CP) after first mention of cognitive performance.
Reply:C.T.A.
Figure 3: Indicate in the figure legend that this is the unweighted polygenic score.
Reply:Done.
Lines 162-169: Here you describe the correlations of the different polygenic scores with each other and with national IQ. This is indeed important, but somewhere in your paper you should place this into a broader perspective by mentioning how many of the 2411 SNPs have highest frequency and how many have the lowest frequency in Europe, Africa, East Asia and South Asia. And you should report how many of the individual SNPs have positive correlations with national IQ, and how many have negative correlations (and perhaps, how many are virtually uncorrelated with IQ). The same for the restricted set of SNPs that Lee et al. consider good candidates for being causal. This is to give a crude indication of how much variation there is in the species, and how important differences between populations are relative to total genetic variance. For example, if 80% of SNPs have positive and 20% have negative correlations with national IQ, this would indicate that there are very substantial race differences compared to total variation. Things would look different if, for example, 55% of the SNPs have positive correlations and 45% have negative correlations. Such an observation would show that there is huge total variation in the species but very little of this is related to race. Many non-geneticists don’t seem to realize what a huge amount of genetic variation there is in our species, both common polymorphisms with small effects and rare variants with sometimes large effects. If there is an opportunity to educate your readers about this, make use of it!
Reply:Very good observation. However, I see a few issues with this because a lot of SNPs are just randomly correlated to national IQ because of spatial autocorrelation or drift, so the r values won't be zero but are wildly distributed in about -0.6 to +0.6 range even for random SNPs.A better measure would be to show the SNPs which have a very high correlation to IQ, but since we have no objective measure to know what a high correlation is, it would provide little validity. Once could see the percentile of the correlation coefficients of random SNPs to IQ, and compare these to the percentiles for the GWAS hits. However, this would add extra technical details to the paper and give little additional benefit over the Monte Carlo simulation I carried out using the different polygenic scores.
Table 2: You should state in the table heading what the dependent variable is in these regressions and the number of cases, in addition to what has been controlled. This table seems to show 3 regression models. Make horizontal lines to separate them, otherwise readers get confused.
Reply:Done.
Line 205: Acronyms! What is BEB, for example? Better spell out the four population names.
Reply:Done.
Line 215: Are these genome-wide significance values for the SNPs?
Reply:Yes
Line 218: To make it easier for the reader to follow, insert one or two sentences stating that this amounts to a weak tendency for SNPs with better statistical significance to be better predictors of IQ. Also, in this exercise you divide up the SNPs according to statistical significance brackets, which means you have somewhat different numbers of SNPs in each bracket. It would be better to rank the SNPs and then define the brackets with equal numbers of SNPs in each. This is because everything else being equal, a polygenic score computed from a smaller number of SNPs will be less accurate than one from a larger number of SNPs and therefore will produce lower correlations with the dependent variable (national IQ). Same for the procedure described in line 259-263.
Reply:I had checked the number contained in each bracket prior to running the analysis because I am aware of this issue and they were almost identical.
I explain in the discussion that "this amounts to a weak tendency for SNPs with better statistical significance to be better predictors of IQ". Should I explain it here too? Sounds like a not so necessary repetition to me.
Lines 224-227: Sentence line 227 should come first because this is the most fundamental piece of information about the height PGS.
Reply:Moved sentence.
In the second sentence you mention the results of a regression. Is this average height in the country predicted with Fst and polygenic score?
Reply:Yes, added this to text.
What is the number of cases in this regression? Such info is important to help the reader figure out what exactly you did.
Reply:23 populations with average height info (added to text).
You mention a slightly negative correlation between height and Edu polygenic scores. This is a bit surprising because phenotypically at the individual level there is a slight positive correlation between height and Edu/IQ. A case of Simpson’s paradox. You may want to discuss this finding somewhere. Is anything known about genetic overlap between height and Edu/IQ? This might mediate some of the phenotypic correlation at the individual level, for example if some SNPs influence both general growth and brain size.
Reply:Well, I run a within superpop (e.g. between European, African, Asian populations separately) correlation between height and PGS and it was positive! However, too small to report because each group has only about 5 populations! My guess is that the weak correlation can be driven simply by East Asians, so removing East Asians could change the results. However, I am not sure I want to include this analysis in the paper because it's not a paper about height. I used height only to show that GWAS hits can predict other phenotypes and don't necessarily and "magically" produce the EAS>Eur>Afr pattern.
Line 342-349: One specific reason for avoiding the less significant GWAS hits in population-level studies is that the linkage phase may be different in different races, a problem that plays no role when comparing individuals within countries. With weak associations, it is not only more likely that the distance between the GWAS hit and the causal polymorphism is larger and linkage is weaker in the European-origin populations that are represented in the GWASs, but it is also more likely that the linkage phase is different in different races.
Reply:Added this to the discussion.
Line 350-358: You can propose the following more formal procedure for constructing the optimal PGS: Start with the quantile that has the most significant SNPs, and then add quantiles in declining order of genome-wide significance. See whether adding quantiles improves or worsens prediction. Initially, adding quantiles will improve prediction, but after a certain point, adding more quantiles will make prediction worse. At that inflection point you have the optimal PGS.
Reply:Since this is the reviewer's original idea, I cited him/her in the manuscript, by changing the paragraph like so: "The predictive accuracy of the PGS in this study is saturated by the high correlation with population IQ, but methods could be used in other studies to improve PGS construction. Reviewer 2 suggested the following procedure for optimal PGS construction that could be used in future studies: “Start with the quantile that has the most significant SNPs, and then add quantiles in declining order of genome-wide significance. Initially, adding quantiles will improve prediction, but after a certain point, adding more quantiles will make prediction worse. At that inflection point you have the optimal PGS”.
Line 368-372: Linkage decay by itself will only reduce prediction, and will do so to the same extent in all populations that are descended from the original mutant (assuming the causal polymorphism is younger than the ones with which it is linked). What messes things up is that there were serial population bottlenecks where a rare recombinant may have been the only survivor carrying the derived allele. In that case the descendants of this bottlenecked population will no longer have the original linkage phase. For example, if ancestral Europeans but not Africans went through such a bottleneck, the linkage phase may be opposite in Europe and Africa, and what predicts higher IQ in Europe predicts lower IQ in Africa. Therefore the urgent need for GWASs in non-European populations, both to find the causal polymorphisms (assumed to predict in the same direction everywhere) and to improve genomic prediction for non-Europeans. Eventually, we want “color-blind” polygenic scores that are computed from causal polymorphisms including those that are polymorphic in only some human populations.
Reply:By “rare recombinant” I suppose you mean someone with a rare LD pattern so that it carries haplotype, for example AG instead of say AC like the rest of the population. If A is the causal allele, and G is the tag allele at the tag SNP, and C has opposite effect to G, then this would indeed create opposite effects across populations. I added a sentence to the last paragrph outlining in more details the benefits of trans-ethnic GWAS.
